# Absolute Phase Retrieval Using One Coded Pattern and Geometric Constraints of Fringe Projection System

**Xu Yang [1], Chunnian Zeng [1], Jie Luo [1], Yu Lei [1], Bo Tao [2] and Xiangcheng Chen [1,\*]**

[1] School of Automation, Wuhan University of Technology, Wuhan 430070, China; yx_auto@whut.edu.cn (X.Y.); zengchn@whut.edu.cn (C.Z.); luo_jie@whut.edu.cn (J.L.); leiyu9087@whut.edu.cn (Y.L.)

[2] Key Laboratory of Metallurgical Equipment and Control Technology, Ministry of Education, Wuhan University of Science and Technology, Wuhan 430081, China; taoboq@wust.edu.cn

[\*] Correspondence: chenxgcg@ustc.edu; Tel.: +86-139-66733394

**Abstract:** Fringe projection technologies have been widely used for three-dimensional (3D) shape measurement. One of the critical issues is absolute phase recovery, especially for measuring multiple isolated objects. This paper proposes a method for absolute phase retrieval using only one coded pattern. A total of four patterns including one coded pattern and three phase-shift patterns are projected, captured, and processed. The wrapped phase, as well as average intensity and intensity modulation, are calculated from three phase-shift patterns. A code word encrypted into the coded pattern can be calculated using the average intensity and intensity modulation. Based on geometric constraints of fringe projection system, the minimum fringe order map can be created, upon which the fringe order can be calculated from the code word. Compared with the conventional method, the measurement depth range is significantly improved. Finally, the wrapped phase can be unwrapped for absolute phase map. Since only four patterns are required, the proposed method is suitable for real-time measurement. Simulations and experiments have been conducted, and their results have verified the proposed method.

**Keywords:** absolute phase retrieval; phase-shift; fringe order; geometric constraints

## 1. Introduction

Optical 3D measurement plays a pivotal role in all aspects of our lives, such as industrial production, biological medicine, and consumer entertainment [1–5]. Many optical technologies including structured light, stereo vision, and digital fringe projection (DFP) have been exploited to achieve high-density and full-field 3D measurement [6]. Among those technologies, DFP has become the most popular one because of its speed, accuracy, and flexibility [7]. Fourier transform and phase-shift are two main methods applied in the DFP system [8]. The former method only uses one pattern for computing phase map, but the measured surfaces must be rather simple to avoid a spectral overlapping problem. On the other hand, the phase-shift method exploits at least three patterns to compute the phase map pixel-by-pixel, which can achieve higher accuracy and stronger robustness, especially for complex surfaces. However, those two methods can only work out wrapped phases which need to be unwrapped for absolute phase maps.

Ideally, when referring to the neighboring pixels, the wrapped phase can be unwrapped by adding integral multiple of $2\pi$ at each pixel. In reality, however, local shadows, random noises, and isolated objects are very usual occurrences that make the unwrapping phase difficult [9]. Thus, many absolute phase retrieval algorithms have been proposed, which can be divided into two major classes: spatial algorithms and temporal algorithms [7]. The spatial algorithms are

generally used for smooth surfaces, while the temporal algorithms are more suitable for complex surfaces and attract more attention [10]. Research conducted in this field brings forth several typical examples. Chen et al. [11,12] first proposed two-wavelength phase-shift interferometry, and then developed multi-wavelength phase-shift interferometry to enhance the measurement capability. Sansoni et al. [13] combined phase-shift and gray-code into the 3D vision system, which greatly improved the measurement performance. Wang et al. [14] put forward an effective and robust phase-coding method. Zheng et al. [15] improved the phase-coding method for a large number of code words. Chen et al. [16,17] successively developed a quantized phase-coding method and a modified gray-level coding method, which achieved good results when measuring isolated objects. Nevertheless, all the aforementioned methods require three or more extra patterns, which will limit the speed of measurement. To reduce the number of patterns, some researchers have utilized color patterns for 3D measurement [18–20]. However, these methods have always failed for colorful objects. Other researchers have employed more cameras to capture the patterns from different perspectives, such that the multi-view geometric constraints can be used for absolute phase calculation [21–23]. However, the measurement field reduces because of the multiple perspectives, and the cost and complexity of the system increase due to additional cameras [24].

To realize high-speed measurement, An et al. [25] recently proposed a pixel-wise phase unwrapping method with no additional pattern. Based on the geometric constraints of fringe projection system, an artificial phase map $\Phi_{min}$ at the closest depth plane $z_{min}$ is generated, and then the phase unwrapping can be executed by referring to $\Phi_{min}$. Subsequently, a number of algorithms were developed for phase unwrapping based on An's method [26–29]. However, the maximum depth range this method can handle is within $2\pi$ in phase domain. When the object points far away from depth plane $z_{min}$ brings more than $2\pi$ changes, this method is no longer applicable.

Inspired by An's method, this paper presents an absolute phase retrieval method using only one additional coded pattern to improve the measurement depth range. Firstly, the wrapped phase is calculated from three phase-shift patterns, and the code word is extracted from the coded pattern. Secondly, an artificial fringe order map $k_{min}$ of depth plane $z_{min}$ is generated, and then the code word is mapped to the fringe order by referring to the fringe order map $k_{min}$. Finally, the wrapped phase is unwrapped for the absolute phase map. Simulations and experiments have been conducted to verify the proposed method.

## 2. Principle

### 2.1. Fringe Projection System

The setup of a typical fringe projection system is shown in Figure 1. This system mainly includes a projector, a camera, and measured objects. The patterns are projected by the projector onto the measured objects from one direction, modulated by the objects' surfaces, and then captured by the camera from another direction. In Figure 1, Points $O_c$ and $O_p$ respectively denote the optical centers of the camera and the projector. The optical axes of the projector and the camera intersect at point $O$ on the reference plane. Note that line $O_cO_p$ is parallel to the reference plane, so points $O_c$ and $O_p$ have the same distance $L$ from the reference surface. Based on the triangulation principle, the height of the measured objects can be computed as [30]:

$$h = \frac{L * \Delta\phi}{2\pi f_0 d + \Delta\phi} \tag{1}$$

where $\Delta\phi$ denotes the phase difference between the point $P$ on the object and the point $B$ on the reference plane, $f_0$ denotes the frequency of the fringe on the reference plane. For a specific system, parameters $L$, $d_0$ and $f_0$ are fixed, which can be obtained by calibration [31].

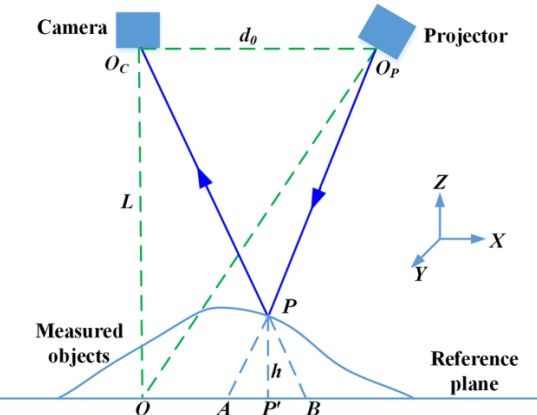

**Figure 1.** Fringe projection system.

### 2.2. Phase-Shift and Coded Patterns

Phase-shift methods have been widely used for optical measurement because of their measurement accuracy, spatial resolution, and data density [8]. The three-step phase-shift method requires the fewest number of patterns among various phase-shift methods, thus it is desirable for high-speed applications. Three-step phase-shift patterns can be described as:

$$
\begin{cases}
I_1(x,y) = A(x,y) + B(x,y)\cos[\phi(x,y) - 2\pi/3] \\
I_2(x,y) = A(x,y) + B(x,y)\cos[\phi(x,y)] \\
I_3(x,y) = A(x,y) + B(x,y)\cos[\phi(x,y) + 2\pi/3]
\end{cases}
\tag{2}
$$

where $A(x,y)$ denotes the average intensity, $B(x,y)$ denotes the intensity modulation, and $\phi(x,y)$ denotes the phase to be solved for. Figure 2a–c shows three phase-shift patterns generated using the above equations, and same rows of the three patterns are shown in Figure 3a. Solving the above equations, the three variables can be calculated as:

$$
\begin{cases}
A(x,y) = (I_1 + I_2 + I_3)/3 \\
B(x,y) = [(I_1 - I_3)^2/3 + (2I_2 - I_1 - I_3)^2/9]^{1/2} \\
\phi(x,y) = \tan^{-1}[\sqrt{3}(I_1 - I_3)/(2I_2 - I_1 - I_3)]
\end{cases}
\tag{3}
$$

Because of the arctangent operation, the wrapped phase $\phi(x,y)$ is limited in range of $[-\pi, \pi]$. Thus, phase unwrapping should be carried out to recover the absolute phase. If the fringe order $k(x,y)$ is determined, the absolute phase $\Phi(x,y)$ can be calculated as:

$$
\Phi(x,y) = \phi(x,y) + 2\pi * k(x,y)
\tag{4}
$$

To determine the fringe order, we designed one coded pattern. Figure 2d shows the coded pattern, and one row of this pattern is shown in Figure 3b. The coded pattern can be described as:

$$
I_M(x,y) = A(x,y) + B(x,y) * M(x,y) = A(x,y) + B(x,y) * [2 * \mathrm{mod}(\lceil x/P \rceil, N)/N - 1]
\tag{5}
$$

where $P$ represents the fringe period, the truncated integer $k = \lceil x/P \rceil$ represents the fringe order, and the remainder $C = \mathrm{mod}(k, N)$ represents the code word; note that it is a periodic function with a period of $N$. Once these four patterns are captured, the coded coefficient $M(x,y)$ ranging from $-1$ to $1$ can be calculated as:

$$
M(x,y) = \cos^{-1}[(I_m - A)/B]
\tag{6}
$$

Then the code word $C(x,y)$ can be computed as:

$$C(x,y) = round[(M+1)*N/2] \tag{7}$$

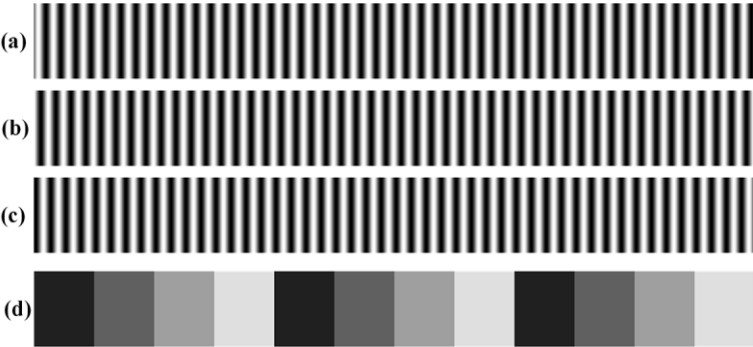

**Figure 2.** Projected patterns. (**a**–**c**) phase-shift patterns; (**d**) coded pattern.

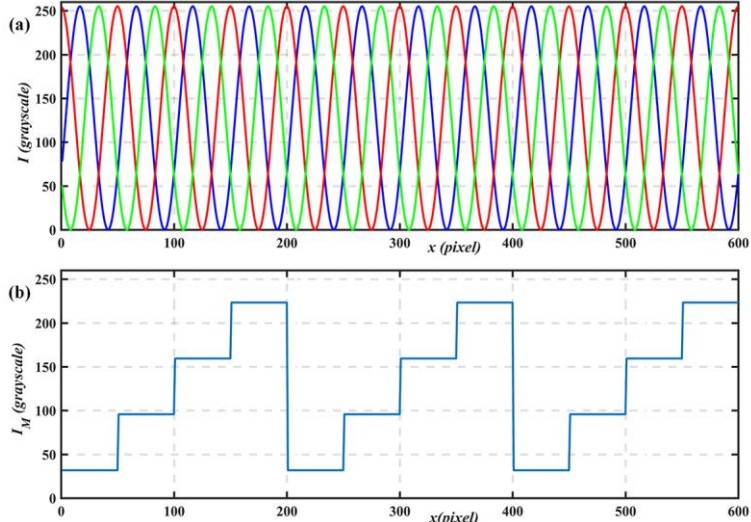

**Figure 3.** Same rows as in Figure 2. (**a**) Phase-shift patterns; (**b**) coded pattern.

### 2.3. Geometric Constraints for Phase Unwrapping

An et al. [25] have recently proposed a pixel-wise phase unwrapping method based on geometric constraints of the fringe projection system. The main idea is to create the minimum phase map $\Phi_{min}$ at the closest depth plane $z_{min}$ of the measured volume. Then phase unwrapping can be performed with reference to minimum phase map $\Phi_{min}$. The details of this method have been described in [25]. The following briefly introduces the main idea of this method.

Figure 4 illustrates the phase unwrapping method using the minimum phase map $\Phi_{min}$. If the wrapped phase $\phi$ is less than $\Phi_{min}$, we need to add $k$ times of $2\pi$ to the wrapped phase $\phi$ to obtain the absolute phase $\Phi$. The fringe order $k$ can be computed as:

$$k(x,y) = ceil\left(\frac{\Phi_{min} - \phi}{2\pi}\right) \tag{8}$$

where function *ceil*() returns the closest upper integer value. It should be noted that the above equation must satisfy the following condition:

$$0 \le \Phi - \Phi_{min} < 2\pi \tag{9}$$

Its physics signification is that the measured objects should be close to the depth plane $z_{min}$ and within $2\pi$ in phase domain. In other words, the maximum depth range should be less than $2\pi$ changes

which will limit the applications of this method. For example, at point A,$\Phi - \Phi_{min} < 2\pi$, and wrapped phase $\phi$ is correctly unwrapped for the absolute phase $\Phi' = \Phi$; at point B, $\Phi - \Phi_{min} > 2\pi$, but wrapped phase $\phi$ is wrongly unwrapped for the absolute phase $\Phi' \neq \Phi$.

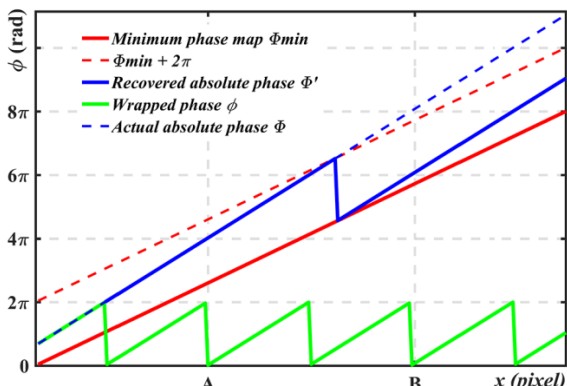

**Figure 4.** Phase unwrapping using the minimum phase map $\Phi_{min}$.

### 2.4. Phase Unwrapping with One Coded Pattern

To improve the measurement depth range, we utilized an additional coded pattern to provide more information for fringe order determination. Assume that the camera captures an object placed at the depth plane $z_{min}$, there exists a one-to-one mapping between the camera sensor and the projector sensor, and the minimum fringe order $k_{min}$ can be uniquely defined on the projector sensor. Figure 5 illustrates the main idea to determine the fringe order $k$, in which line $k_{min}$ plots the minimum fringe order, the line $C$ plots the code word at depth plane $z$, and line $k$ plots the corresponding fringe order. The relationship between the three variables can be described as:

$$k = C + N * ceil\left(\frac{k_{min} - C}{N}\right) \tag{10}$$

For example, at point D, $k_{min} - C < 0$, thus $k = C$; at point E, $0 < k_{min} - C < N$, thus $k = C + N$; at point F, $N < k_{min} - C < 2 * N$, thus $k = C + 2 * N$. Similarly, the above equation must satisfy the following condition:

$$0 \leq k - k_{min} < N \tag{11}$$

In other words, the measured objects should be close to the depth plane $z_{min}$ within $2\pi N$ in phase domain. Through the above analysis, the proposed method raises the measurement depth range by $N$ times compared with the traditional method.

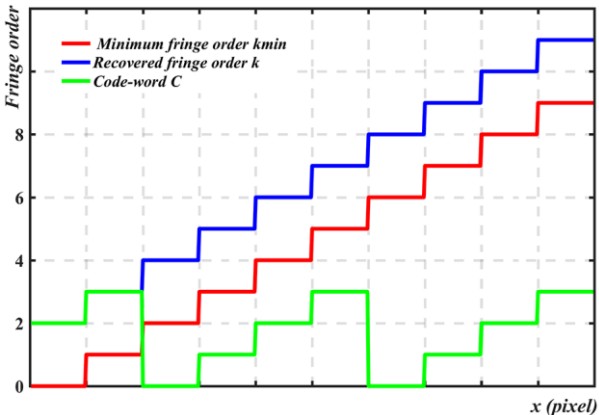

**Figure 5.** Fringe order determination using the minimum fringe order $k_{min}$.

## 3. Simulation

To test the performance of the proposed method, some simulations were carried out. Figure 6 shows the simulation of the closet depth plane $z_{min}$. Specifically, Figure 6a–c shows three phase-shift patterns with eight periods; Figure 6d shows the corresponding wrapped phase ranging from $-\pi$ to $\pi$; Figure 6e shows the fringe order map regarded as $k_{min}$; and Figure 6f shows the absolute phase map regarded as $\Phi_{min}$.

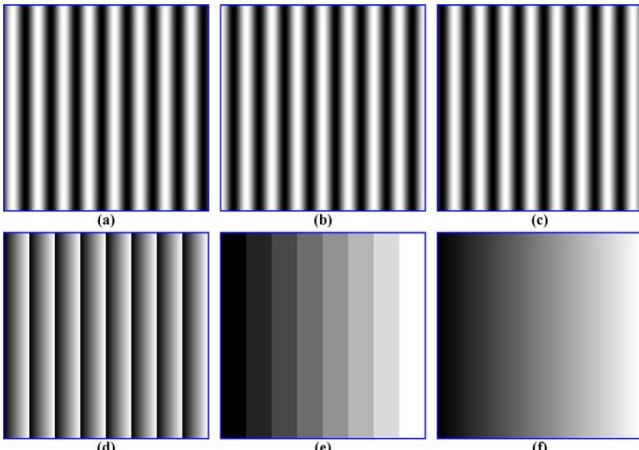

**Figure 6.** Simulation of depth plane $z_{min}$. (**a**–**c**) Phase-shift patterns; (**d**) wrapped phase map; (**e**) minimum fringe order map $k_{min}$; (**f**) minimum phase map $\Phi_{min}$.

Then, a hemisphere was selected as the measure object and simulated, as shown in Figure 7. Specifically, Figure 7a–c shows the three phase-shift patterns; Figure 7d shows the coded pattern with $N = 4$; Figure 7e shows the fringe order determined by the proposed method; Figure 7f shows the fringe order map determined by An's method for comparison; Figure 7g shows the absolute phase map recovered by the proposed method; Figure 7h shows the absolute phase map recovered by An's method. Obviously, the fringe order and the absolute phase map are correctly determined by the proposed method. However, An's method fails in contrast. The 3D reconstruction results of the hemisphere using the two methods are shown in Figure 8. As we can see, the proposed method can accurately recover the whole surface of the hemisphere, but An's method fails to measure the overall hemisphere surface. The maximum depth range of the proposed method can deal with is $2\pi N$, which is four times that of An's method.

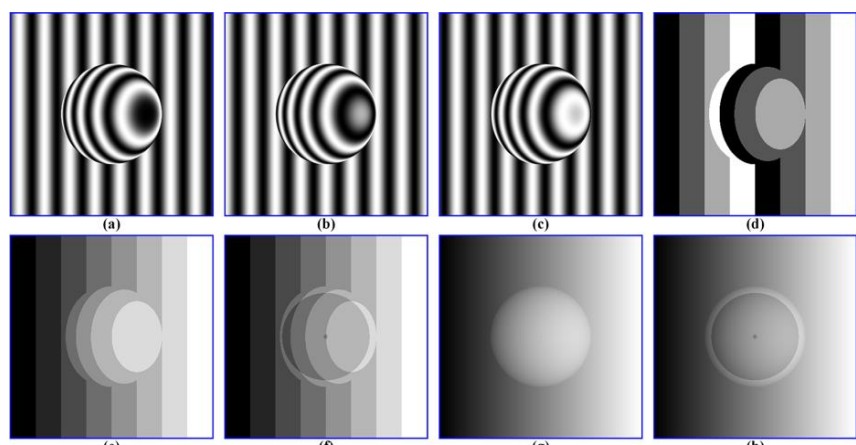

**Figure 7.** Simulation of a hemisphere. (**a**–**c**) Phase-shift patterns; (**d**) coded pattern; (**e**) fringe order map using the proposed method; (**f**) fringe order map using An's method; (**g**) absolute phase map using the proposed method; (**h**) absolute phase map using An's method.

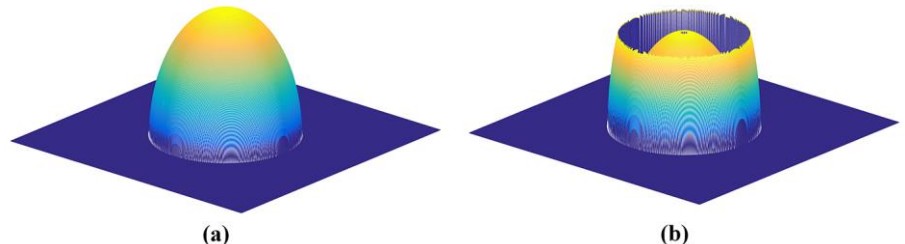

**Figure 8.** A 3D reconstruction of the hemisphere. (**a**) The proposed method; (**b**) An's method.

## 4. Experimental Setup

To test the proposed method in real condition, a fringe projection system was set up. The system consisted of a projector (Light Crafter 4500) with resolution of 912 × 1140 pixels, and a camera (IOI Flare 2M360-CL) with resolution of 1280 × 1024 pixels. A flat board was placed at the closest depth plane of the measured volume, and used as the reference plane. Two isolated objects were selected as the measured objects. Total four patterns, including three phase-shift patterns and one coded pattern, were projected onto the reference plane and the measured objects by the projector, and sequentially captured by the camera.

Figure 9a–c shows three phase-shift patterns projected onto the reference plane, respectively. Figure 9d shows the corresponding wrapped phase. Figure 9e shows the fringe order, also regarded as the minimum fringe order map $k_{min}$. Figure 9f shows the absolute phase map also regarded as the minimum phase map $\Phi_{min}$. Similarly, Figure 10a–c shows the images of three phase-shift patterns projected onto the measured objects, respectively. Figure 10d shows the corresponding wrapped phase map calculated from the three phase-shift patterns. Meanwhile, the average intensity and intensity modulation were calculated. Figure 10e shows the coded pattern with $N = 4$, and Figure 10f shows the corresponding code word map.

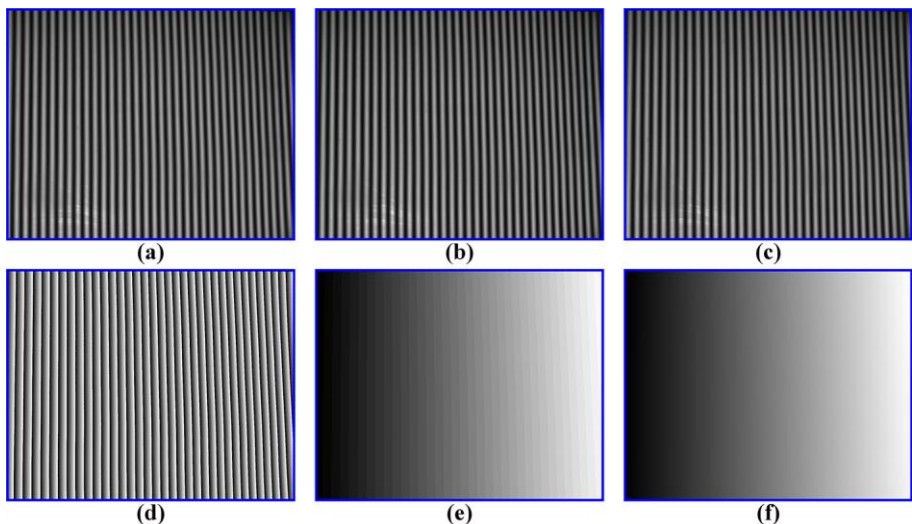

**Figure 9.** Images of the reference plane. (**a–c**) Phase-shift patterns; (**d**) wrapped phase map; (**e**) minimum fringe order map $k_{min}$; (**f**) minimum phase map $\Phi_{min}$.

In order to compare the proposed method and An's method, Equations (12) and (14) were both used for computing fringe order. Figure 11a,b shows the fringe order maps recovered by the two methods. As we can see, the proposed method recovered the fringe order map $\Phi$ correctly; however, An's method led to the wrong fringe order map $\Phi'$ at some areas. There are obvious differences between the two fringe order maps within the two circular areas plotted in Figure 11. The pixels of the

same stripe had the same fringe order *k* in Figure 11a. However, the pixels of the same stripe had a different fringe order *k′* in Figure 11b.

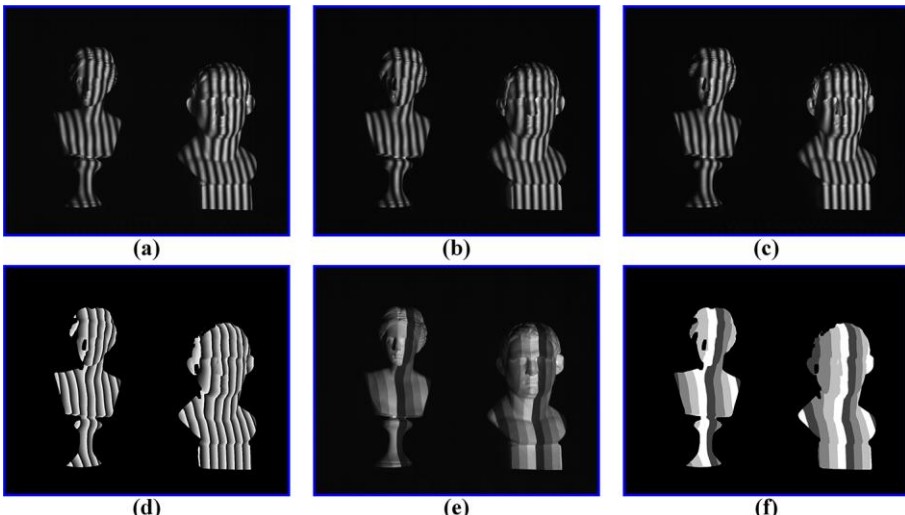

**Figure 10.** Images of the measured objects. (**a**–**c**) Phase-shift patterns; (**d**) wrapped phase map; (**e**) coded pattern; (**f**) code-word map.

For better illustration, Figure 12a,b shows the 600th rows of the two fringe order maps and absolute phase maps. Clearly, $\Phi - \Phi_{min} < 8\pi$ and $\Phi' - \Phi_{min} < 2\pi$. This indicates that the maximum depth range of the proposed method is up to $8\pi$, and that of An's method is only $2\pi$. Therefore, the proposed method can obtain much larger depth range than An's method. Finally, we reconstructed the 3D shapes of the two isolated objects, as shown in Figure 13.

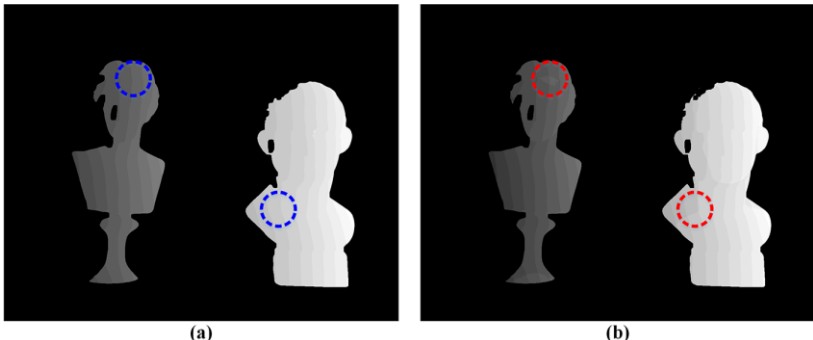

**Figure 11.** Fringe order maps. (**a**) The proposed method; (**b**) An's method.

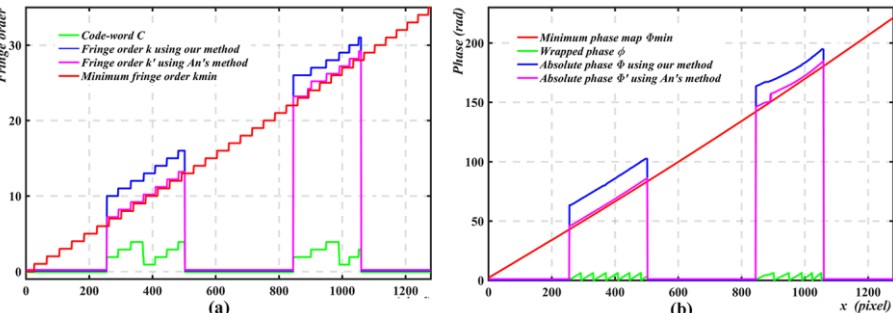

**Figure 12.** The 600th rows. (**a**) Fringe order maps; (**b**) absolute phase maps.

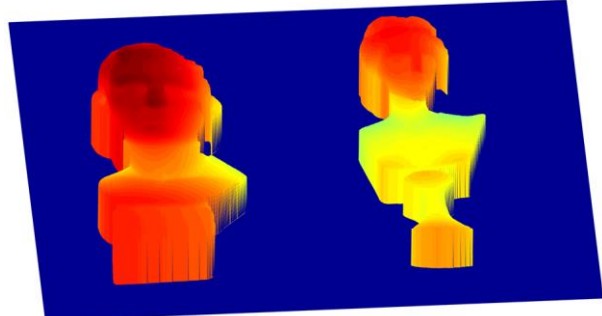

**Figure 13.** Measurement result of two isolated objects.

In order to further verify our method, two separate planes were also measured using the proposed method. Figure 14a–c shows three phase-shift patterns projected onto the two planes, respectively. Figure 14d shows the corresponding wrapped phase map. Figure 14e shows the coded pattern, and Figure 14f shows the corresponding code-word map. Then the fringe order was calculated, as shown in Figure 15a. Using Equation (4), the absolute phase map was recovered, as shown in Figure 15b. Finally, the 3D shapes of two planes were reconstructed, as shown in Figure 16. There are no obvious mistakes in the measurement results. The experimental results illustrate the performance of the proposed method.

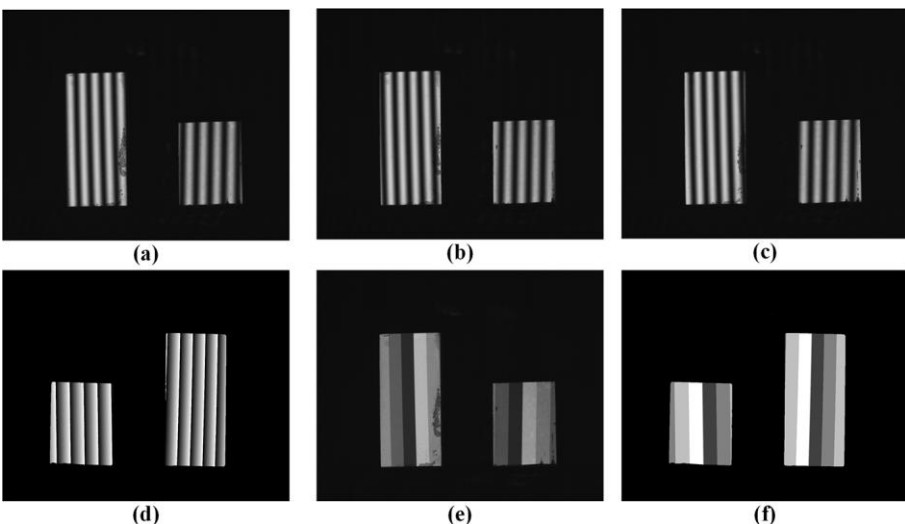

**Figure 14.** Images of two planes. (**a**–**c**) Phase-shift patterns; (**d**) wrapped phase map; (**e**) coded pattern; (**f**) code-word map.

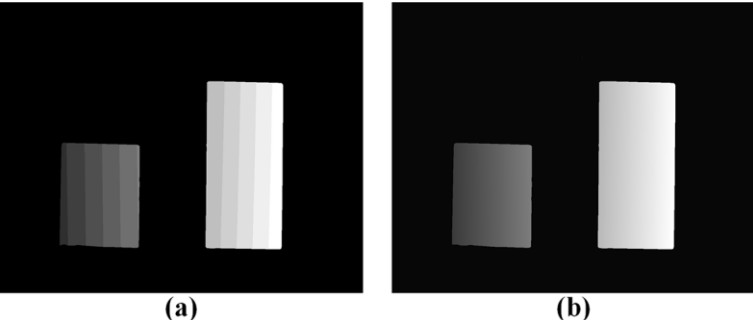

**Figure 15.** (**a**) Fringe order map; (**b**) absolute phase map.

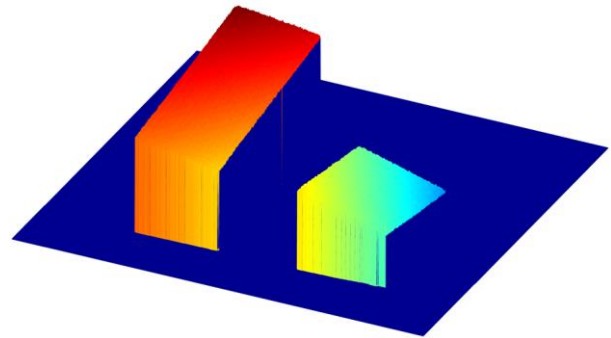

**Figure 16.** Measurement result of two planes.

## 5. Conclusions

This paper has presented an absolute phase retrieval method using only one coded pattern. A total of four patterns are used for 3D shape measurement, which is suitable for high-speed applications. The code words are encoded into the coded pattern, which can be correctly recovered using the average intensity and intensity modulation of phase-shift patterns. Based on the geometric constraints of fringe projection system, the minimum fringe order map is generated, then the code word can be easily converted into fringe order. Compared with the conventional method, the proposed method can significantly enhance the measurement depth range.

**Author Contributions:** X.C. and B.T. conceived and designed the experiments; X.Y. and J.L. performed the experiments; X.C. and C.Z. analyzed the data; X.Y. and Y.L. wrote the paper.

**Funding:** This research was funded by National Natural Science Foundation of China (51605130), Hubei Provincial Natural Science Foundation of China (2018CFB656), Fundamental Research Funds for the Central Universities (WUT: 2017IVA059), Open Fund of the Key Laboratory for Metallurgical Equipment and Control of Ministry of Education in Wuhan University of Science and Technology (2018B03).

**Conflicts of Interest:** The authors declare no conflicts of interest.

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
