# Peer review of "Absolute Phase Retrieval Using One Coded Pattern and Geometric Constraints of Fringe Projection System"

_applsci, doi:10.3390/app8122673_

Reviewer 1 Report

The paper faces the problem of the 3-D shape measurement. More specifically, it deals with fringe projection technologies. In such a scenario, absolute phase recovery is crucial. Thus, an absolute phase retrieval method is proposed with only one coded pattern. It is shown that such a method can be suitable for real-time 3D shape measurement. Finally, experimental results validate the effectiveness of the proposed solution.
I think that this work is solid. Section 2. in particular is very appreciable, because it exemplifies the problem. Section 3. show good results, but, in my opinion, more details should be provided about the methodology adopted for the experiments.
Overall, the paper is well-written, but the background should be completed with some works which deal with the more general problem of signal reconstruction in multidimensional spaces (e.g. [1-3]).
Moreover, stochastic sampling should be brieafly recalled [4-6]. I see that, for the particular application studied in this paper, the sampling theory and the point processes theory cannot be directly applied, but the reader would take advantage of a wider introduction (now, it seems to discourage  a reader which is not an expert of the very specific topic).

MINORS

Equations 2,3, and 4 should become 2a,2b, and 2c, respectively. (three components of the same vector). I suggest the same also for eq. 5-7 (3a,3b, 3c).

BIBLIOGRAPHY

[1] “Distributed field reconstruction in wireless sensor networks based on hybrid shift-invariant spaces,” IEEE Trans. Signal Process., vol. 60, no. 10, pp. 5426–5439, Oct 2012.
[2] “Environmental monitoring systems: A review,” IEEE Sensors Journal, vol. 13, no. 4, pp. 1329– 1339, April 2013
[3] “On Landau’s necessary density conditions for sampling and interpolation of band-limited functions,”J. London Math. Soc., vol. 54, no. 3, pp. 557–565, 1996.
[4] “Ginibre sampling and signal reconstruction,” in 2016 IEEE International Symposium on Information Theory (ISIT), July 2016, pp. 865–869."
[5] “On random sampling with nodes attraction: The case of Gauss-Poisson process,” in 2017 IEEE International Symposium on Information Theory (ISIT), June 2017, pp. 2278–2282.
[6] "Observer-based output feedback control for sampled-data system with Markovian packet losses and stochastic sampling", IECON 2017 - 43rd Annual Conference of the IEEE Industrial Electronics Society Year: 2017, Page s: 4319 - 4324

Author Response

Reply to Reviewer 1

Manuscript ID: applsci-389519    

Title: Absolute phase retrieval using one coded pattern and geometric constraints of fringe projection system

Dear Sir/Madam

Thank you very much for reviewing our paper and giving invaluable suggestions on how to improve it. Below are our replies to your comments for your kind approval please, and we have revised the paper accordingly.

Yours sincerely

Xu Yang , Chunnian Zeng , Jie Luo , Yu Lei , Bo Tao , Xiangcheng Chen *

Overall comment: The paper faces the problem of the 3-D shape measurement. More specifically, it deals with fringe projection technologies. In such a scenario, absolute phase recovery is crucial. Thus, an absolute phase retrieval method is proposed with only one coded pattern. It is shown that such a method can be suitable for real-time 3D shape measurement. Finally, experimental results validate the effectiveness of the proposed solution.

Response: We are grateful to the reviewer’s comments and suggestions. We would like to assure the reviewer that we have addressed all the issues raised by the reviewer in the revised paper. The specific replies to the reviewer’s comments and suggestions are given below.

Comment 1:  I think that this work is solid. Section 2. in particular is very appreciable, because it exemplifies the problem. Section 3. show good results, but, in my opinion, more details should be provided about the methodology adopted for the experiments.

Response: Thanks very much for your comment. In order to better verify the proposed method. We carried out some simulations and more experiments.  Section 3 introduces the simulations in detail. The closet depth plane zmin and a hemisphere was simulated. Then we use the proposed method and An’s method to determine the fringe order and recover the absolute phase map. From the 3D reconstruction results using the two methods in Figure 8, from which we can see that the proposed method works well, however An’s method fails. Section 4 adds an experiment of two separate planes. And we use the proposed method to measure them. Figure 16 shows the reconstructed 3D shapes of the two planes with no obvious errors. Both the simulations and experiments illustrate the performance of the proposed method. More details can be found in Section 3 and Section 4.

Figure 8.  3D reconstruction of the hemisphere. (a) The proposed method; (b) An’s method.

Figure 16.  Measurement result of two planes.

Comment 2: Overall, the paper is well-written, but the background should be completed with some works which deal with the more general problem of signal reconstruction in multidimensional spaces (e.g. [1-3]). Moreover, stochastic sampling should be briefly recalled [4-6]. I see that, for the particular application studied in this paper, the sampling theory and the point processes theory cannot be directly applied, but the reader would take advantage of a wider introduction (now, it seems to discourage  a reader which is not an expert of the very specific topic).

BIBLIOGRAPHY

[1] “Distributed field reconstruction in wireless sensor networks based on hybrid shift-invariant spaces,” IEEE Trans. Signal Process., vol. 60, no. 10, pp. 5426–5439, Oct 2012.

[2] “Environmental monitoring systems: A review,” IEEE Sensors Journal, vol. 13, no. 4, pp. 1329– 1339, April 2013

[3] “On Landau’s necessary density conditions for sampling and interpolation of band-limited functions,”J. London Math. Soc., vol. 54, no. 3, pp. 557–565, 1996.

[4] “Ginibre sampling and signal reconstruction,” in 2016 IEEE International Symposium on Information Theory (ISIT), July 2016, pp. 865–869."

[5] “On random sampling with nodes attraction: The case of Gauss-Poisson process,” in 2017 IEEE International Symposium on Information Theory (ISIT), June 2017, pp. 2278–2282.

[6] "Observer-based output feedback control for sampled-data system with Markovian packet losses and stochastic sampling", IECON 2017 - 43rd Annual Conference of the IEEE Industrial Electronics Society Year: 2017, Page s: 4319 - 4324

Response:  Thanks very much for your comments. We have carefully read the six papers that you provided for us. These papers have inspired us a lot. We are very sorry that we have not reviewed these papers. Since the main idea of our paper is about 3D shape measurement, which has a relatively little correlation with the signal sampling and reconstruction. Nevertheless, another paper in writing called “Narrow laser echo signal processing for lidar based on transmission pulse spacing modulation” will also be submitted to Applied Sciencesrecently, adopts the equivalent sampling theory similar to the stochastic sampling in Ref[4-5]. In that paper, we will cite all the reference about signal reconstruction in Ref[1-6]. Hope you can understand and thank you very much again.

Comment 3: Equations 2,3, and 4 should become 2a,2b, and 2c, respectively. (three components of the same vector). I suggest the same also for eq. 5-7 (3a,3b, 3c).

Response: Thanks for your comment. Doing accordingly, Equations 2,3, and 4 are combined into Equation 2 in the revised paper. Similarly, Equations 5,6, and 7 are combined into Equation 3 in the revised paper.

Reviewer 2 Report

 The present paper reports an absolute phase retrieval method with a fringe projection system. The topic is of interest to the community of optical metrology. I think that it suits very well the applied sciences scheme.

Author Response

Reply to Reviewer 2

Manuscript ID: applsci-389519     

Title: Absolute phase retrieval using one coded pattern and geometric constraints of fringe projection system

Dear Sir/Madam

Thank you very much for reviewing our paper and giving invaluable suggestions on how to improve it. Below are our replies to your comments for your kind approval please, and we have revised the paper accordingly.

Yours sincerely

Xu Yang , Chunnian Zeng , Jie Luo , Yu Lei , Bo Tao , Xiangcheng Chen *

Overall comment :  The present paper reports an absolute phase retrieval method with a fringe projection system. The topic is of interest to the community of optical metrology. I think that it suits very well the applied sciences scheme.

Response: We are grateful to the reviewer’s comments and suggestions. Thanks very much.

Reviewer 3 Report

This paper proposed new phase unwrapping algorithms using one coded pattern and geometric constraints.  In general, the article is easy to follow. The method is also interesting.  But a few comments need to be addressed:

(1)  the In introduction, authors have reviewed a few papers (including high speed 3D measurement), but have not really clarified why this work is necessary. It means that it is difficult to see how much significant differences between the current paper with previous papers.

(2)  In eq1, there an is angle, but In Figure 1, the angle is missing.  

(3)  It is nice that authors are able to pickup one method (ref 15) and dive into details. But I have not seen why the method is particularly wrong only based on Figure 4.

(4)  Author’s extension is rather a progressive development based on the existing method (not quite a breakthrough yet). Authors may want to find more significant innovation for their work.

(5)  More experimental data needed to prove the robustness of the proposed method.

(6)  For results, are there any ground truth data? It maybe better to use ground truth simulation data to prove the effectiveness of the proposed method. 

Author Response

Reply to Reviewer 3

Manuscript ID: applsci-389519     

Title: Absolute phase retrieval using one coded pattern and geometric constraints of fringe projection system

Dear Sir/Madam

Thank you very much for reviewing our paper and giving invaluable suggestions on how to improve it. Below are our replies to your comments for your kind approval please, and we have revised the paper accordingly.

Yours sincerely

Xu Yang , Chunnian Zeng , Jie Luo , Yu Lei , Bo Tao , Xiangcheng Chen *

Overall comment :  This paper proposed new phase unwrapping algorithms using one coded pattern and geometric constraints.  In general, the article is easy to follow. The method is also interesting.  But a few comments need to be addressed.

Response: We are grateful to the reviewer’s comments and suggestions. We would like to assure the reviewer that we have addressed all the issues raised by the reviewer in the revised paper. The specific replies to the reviewer’s comments and suggestions are given below.

Comment 1: In the introduction, authors have reviewed a few papers (including high speed 3D measurement), but have not really clarified why this work is necessary. It means that it is difficult to see how much significant differences between the current paper with previous paper.

Response: Thanks very much for your comment. More papers have been reviewed in the revised paper, including some reviews [3-5], and some research articles [18-24].

Comment 2: In eq1, there an is angle, but in Figure 1, the angle is missing. 

Response:  Thanks very much for your comment. The angle Δϕ denotes the phase difference between the point P on the object and the point B on the reference plane. The phase ϕ is carried by the projected patterns, and these patterns are described in Equation 2. Using Equation 3 and suitable phase unwrapping method, the phase ϕ of the object and the reference plane can be calculated, then the phase difference Δϕ can be obtained. Therefore, the angle Δϕ is not an angle of the physical structure, thus not shown in Figure 1. A similar figure of the fringe projection system can be found in ref[30].

Comment 3:   It is nice that authors are able to pick up one method (ref 15) and dive into details. But I have not seen why the method is particularly wrong only based on Figure 4. 

Response:  Thanks very much for your careful work. We are very sorry for the mark error of the blue solid line in the original paper.  In the revised Figure 4,  we have corrected the error. The blue dot line Ф denotes the actual absolute phase; the blue solid line Фdenotes the absolute phase using An’s method.  At point A, , An’s method works well since . However, at point B, , An’s method fails.

Figure 4. Phase unwrapping using the minimum phase map Фmin.

Comment 4: Author’s extension is rather a progressive development based on the existing method (not quite a breakthrough yet). Authors may want to find more significant innovation for their work.

Response:  Thanks very much for your comment. General phase retrieval methods requires three or more coded patterns, such as gray-code[13] and phase-coding[14] methods. The proposed method only uses one coded pattern. Thus total four patterns are used for 3D shape measurement, which is suitable for high-speed applications. With help of the average intensity and intensity modulation of phase-shift patterns, the code-word can be recovered correctly. Compared with the An’s method, the proposed method can significantly enhance the measurement depth range, which improves the scope of applications of geometric constraints.

Comment 5 and Comment 6: More experimental data needed to prove the robustness of the proposed method. For results, are there any ground truth data? It may be better to use ground truth simulation data to prove the effectiveness of the proposed method.

Response:  Thanks very much for your comment. In order to better verify the proposed method. We carried out some simulations and more experiments.  Section 3 introduces the simulations in detail. The closet depth plane zmin and a hemisphere was simulated. Then we use the proposed method and An’s method to determine the fringe order and recover the absolute phase map. From the 3D reconstruction results using the two methods in Figure 8, from which we can see that the proposed method works well, however An’s method fails. Section 4 adds an experiment of two separate planes. And we use the proposed method to measure them. Figure 16 shows the reconstructed 3D shapes of the two planes with no obvious errors. Both the simulations and experiments illustrate the performance of the proposed method. More details can be found in Section 3 and Section 4.

Figure 8.  3D reconstruction of the hemisphere. (a) The proposed method; (b) An’s method.

Figure 16.  Measurement result of two planes.

Round  2

Reviewer 3 Report

Thanks for the author's work. My previous comments have been addressed.

(1)  Ln 144: 2.4. The Proposed Method is confusing. It should be given the actual name of the method

(2) Figure 4, 5, and 12 should have legend indicating the meaning of different colored lines.

(3) Figure 8, An'method yielded the wrong results. But please give more detailed comments for why An's method failed to achieve desired results.

(4) Lin 191, 4. Experiment should be Experimental setup.

Author Response

Reply to Reviewer 3

Manuscript ID: applsci-389519     

Title: Absolute phase retrieval using one coded pattern and geometric constraints of fringe projection system

Dear Sir/Madam

Thank you very much for reviewing our paper and giving invaluable suggestions on how to improve it. Below are our replies to your comments for your kind approval please, and we have revised the paper accordingly.

Yours sincerely

Xu Yang , Chunnian Zeng , Jie Luo , Yu Lei , Bo Tao , Xiangcheng Chen *

Overall comment :  Thanks for the author's work. My previous comments have been addressed.

Response: We are grateful to the reviewer’s comments and suggestions. We would like to assure the reviewer that we have addressed all the issues raised by the reviewer in the revised paper. The specific replies to the reviewer’s comments and suggestions are given below.

Comment 1:   Ln 144: 2.4. The Proposed Method is confusing. It should be given the actual name of the method.

Response: Thanks for your comment. According this comment, we have revised “ The Proposed Method”  for “ Phase Unwrapping with One Coded Pattern”.

Comment 2: Figure 4, 5, and 12 should have legend indicating the meaning of different colored lines.

Response:  Thanks for your comment. According this comment, we have added legend for Figure 4, 5, and 12. And the revised figures are shown below.

Figure 4. Phase unwrapping using the minimum phase map Фmin.

Figure 5. Fringe order determination using the minimum fringe order kmin.

Figure 12. The 600th rows. (a) Fringe order maps; (b) Absolute phase maps.

Comment 3: Figure 8, An's method yielded the wrong results. But please give more detailed comments for why An's method failed to achieve desired results.

Response:   Thanks for your comment.  According this comment, we have added some comments on Figure 8 in the revised paper. The comment is given below.

“As we can see, the proposed method can accurately recover the whole surface of the hemisphere, but An’s method fails to measure the overall hemisphere surface. The maximum depth range the proposed method can deal with is 2πN, which is four times of An’s method.”

Comment 4: Lin 191, 4. Experiment should be Experimental setup.

Response:  Thanks for your comment. According this comment, we have revised “ Experiment”  for “ Experimental Setup”.
